# Acetabular Coverage Area Occupied by the Femoral Head as an Indicator of Hip Congruency

**DOI:** 10.3390/ani12172201

**Published:** 2022-08-26

**Authors:** Pedro Franco-Gonçalo, Diogo Moreira da Silva, Pedro Leite, Sofia Alves-Pimenta, Bruno Colaço, Manuel Ferreira, Lio Gonçalves, Vítor Filipe, Fintan McEvoy, Mário Ginja

**Affiliations:** 1Department of Veterinary Science, University of Trás-os-Montes and Alto Douro (UTAD), 5000-801 Vila Real, Portugal; 2Veterinary and Animal Research Centre (CECAV), University of Trás-os-Montes and Alto Douro (UTAD), 5000-801 Vila Real, Portugal; 3Associate Laboratory for Animal and Veterinary Sciences (AL4AnimalS), 5000-801 Vila Real, Portugal; 4School of Science and Technology, University of Trás-os-Montes and Alto Douro (UTAD), 5000-801 Vila Real, Portugal; 5Neadvance Machine Vision SA, 4705-002 Braga, Portugal; 6Department of Animal Science, University of Trás-os-Montes and Alto Douro (UTAD), 5000-801 Vila Real, Portugal; 7Department of Engineering, University of Trás-os-Montes and Alto Douro (UTAD), 5000-801 Vila Real, Portugal; 8Institute for Systems and Computer Engineering (INESC-TEC), Technology and Science, 4200-465 Porto, Portugal; 9Department of Veterinary Clinical Sciences, Faculty of Health and Medical Sciences, University of Copenhagen, 1165 Copenhagen, Denmark

**Keywords:** hip dysplasia, FCI scoring, congruency, dog

## Abstract

**Simple Summary:**

Radiographic diagnosis is essential for the genetic control of canine hip dysplasia (HD). The *Fédération Cynologique Internationale* (FCI) scoring HD scheme is based on objective and qualitative radiographic criteria. Subjective interpretations can lead to errors in diagnosis and, consequently, to incorrect selective breeding, which in turn impacts the gene pool of dog breeds. The aim of this study was to use a computer method to calculate the Hip Congruency Index (HCI) to objectively estimate radiographic hip congruency for future application in the development of computer vision models capable of classifying canine HD. The HCI measures the percentage of acetabular coverage that is occupied by the femoral head. Normal hips are associated with an even, parallel joint surface that translates into reduced acetabular free space, which increases with hip subluxation and becomes maximal in hip dislocation. We found statistically significant differences in mean HCI values among all five FCI categories. These results demonstrate that the HCI reliably reflects the different degrees of congruency associated with HD. Therefore, it is expected that when used in conjunction with other HD evaluation parameters, such as Norberg angle and assessment of osteoarthritic signs, it can improve the diagnosis by making it more accurate and unequivocal.

**Abstract:**

Accurate radiographic screening evaluation is essential in the genetic control of canine HD, however, the qualitative assessment of hip congruency introduces some subjectivity, leading to excessive variability in scoring. The main objective of this work was to validate a method-Hip Congruency Index (HCI)-capable of objectively measuring the relationship between the acetabulum and the femoral head and associating it with the level of congruency proposed by the *Fédération Cynologique Internationale* (FCI), with the aim of incorporating it into a computer vision model that classifies HD autonomously. A total of 200 dogs (400 hips) were randomly selected for the study. All radiographs were scored in five categories by an experienced examiner according to FCI criteria. Two examiners performed HCI measurements on 25 hip radiographs to study intra- and inter-examiner reliability and agreement. Additionally, each examiner measured HCI on their half of the study sample (100 dogs), and the results were compared between FCI categories. The paired *t*-test and the intraclass correlation coefficient (ICC) showed no evidence of a systematic bias, and there was excellent reliability between the measurements of the two examiners and examiners’ sessions. Hips that were assigned an FCI grade of A (*n* = 120), B (*n* = 157), C (*n* = 68), D (*n* = 38) and E (*n* = 17) had a mean HCI of 0.739 ± 0.044, 0.666 ± 0.052, 0.605 ± 0.055, 0.494 ± 0.070 and 0.374 ± 0.122, respectively (ANOVA, *p* < 0.01). Therefore, these results show that HCI is a parameter capable of estimating hip congruency and has the potential to enrich conventional HD scoring criteria if incorporated into an artificial intelligence algorithm competent in diagnosing HD.

## 1. Introduction

Congruency is a medical term that derives from the Latin *congruentia*. It means “suitable, proper, harmoniously joined or related”, and is directly associated with the direct relationship of an object with the use for which it is intended [1]. A congruent hip joint is represented as an excellent adaptation of the spherical femoral head to the acetabulum, with minimal, and even converging, joint space between the articular surfaces [2].

Occasionally, in dog there is an abnormal development of bone structures on the hip joint, followed by synovial inflammation and articular cartilage damage, resulting in inadequate hip development with different levels of incongruency between the femoral head and the acetabulum. This condition is commonly denominated as hip dysplasia (HD) [3].

Canine HD is an inherited disease influenced by environmental factors that lead to osteoarthritis and various degrees of disability [4]. There is no definitive genetic diagnosis, and by convention the diagnosis still requires radiographic examination. The radiographic diagnosis is essential for both treatment purposes and for breeding programs based on the selection of animals with phenotypically normal hips, aiming to reduce the frequency of defective genes and consequently the prevalence of the disease in the canine population. Canine HD is a disease of global importance; as such, there are several radiographic scoring systems used worldwide. The *Fédération Cynologique Internationale* (FCI) criteria prevail in most continental European countries, the British Veterinary Association/Kennel Club (BVA/KC) criteria are predominant in the United Kingdom and the Orthopedic Foundation for Animals (OFA) criteria regulate screening in the United States of America [4,5]. Only the breeding of animals with the best hips is allowed or recommended, with the acceptable cutoff being defined for each breed, always preserving at least 50% of the population [4]. Canine HD is a biomechanical disorder, particularly prevalent in large and giant breeds of dogs, but can occur in dogs of any size. The Greyhound is one of a few breeds that are rarely affected by canine HD; its functional selection, which requires a balanced growth of the muscle and skeleton, is thought to have been decisive for the development of healthy hip joints [6]. The greater range and versatility of motion in all directions of the hip joint in dogs, cats and humans requires an insufficient acetabular cover of the femoral head, making it very susceptible to the development of HD [7].

The FCI propose a five-grade scoring system to represent the severity of the disease: A (normal), B (almost normal-transition), C (mild HD), D (moderate HD), and E (severe HD). Hip congruency is regarded as a fundamental parameter for the FCI scoring scheme, as it is directly associated with its categories: A (excellent congruency, parallel subchondral bone); B (transition to mild incongruency, diverging femoral head and cranial acetabular margin); C (moderate incongruency, femoral head separated from acetabulum); D (obvious or considerable incongruency) and E (severe incongruency) [8,9]. As a result of relying solely on a qualitative methodology to evaluate both hip congruency and osteoarthritis, there is some subjectivity, and the accuracy of the scoring varies with the training and expertise of the examiner [9]. Studies have shown low intra-observer and inter-observer agreement when it comes to the assessment of hip morphological characteristics and hip scoring according to FCI standards [10,11,12]. Thus, there is a demand for a more effective and reliable means to evaluate HD based on hip congruency and other joint conformation parameters.

Currently, emerging paradigms associated with computer vision and artificial intelligence have transformed the modus operandis of the medical diagnosis industry. Several computer-assisted diagnostic techniques have emerged and allowed the automation of processes, producing more efficient results [13]. In some of these technological solutions, images with specific annotations are used as ground-truth data to train computer vision models. This allows them to successfully identify appropriate anatomical landmarks and subsequently give correct classification in novel images presented to the model after training [14,15,16].

The main aim of this research was to study the relationship between the acetabular coverage area and the femoral head, and its association with the level of hip congruency. For this purpose, we calculated the HCI, an objective parameter that measures the percentage of the acetabular coverage area that is occupied by the femoral head, and compared it to the different FCI categories. To the authors’ knowledge, the HCI has never been used in previous studies, and it seems to us to be a potentially objective and reliable indicator of hip congruency. The index also lends itself to computer vision applications, and so assists in the autonomous identification and classification of canine HD. In addition, the index was developed using computer vision techniques, and thus can help in the automatic identification and classification of canine HD.

## 2. Materials and Methods

This was a retrospective study based on the evaluation of ventrodorsal hip extended (VDHE) views of 200 dogs (400 hip joints) that were randomly selected: 100 dogs from the Veterinary Teaching Hospital of the University of Trás-os-Montes and Alto Douro database and 100 dogs from the Danish Kennel Club database, obtained between 2010 and 2020. Recorded data included breed and sex.

The dogs included in this study consisted of 35 different breeds: Portuguese Pointers (44/200, 22%), Estrela Mountain Dogs (36/200, 18%), Labrador Retrievers (18/200, 9%), German Shepherds (14/200, 7%), Transmontano Mastiffs (14/200, 7%), Golden Retrievers (9/200, 4.5%), Belgian Shepherds (4/200, 2%), Border Collies (3/200, 1.5%), German Wirehaired Pointers (3/200, 1.5%), Rottweilers (3/200, 1.5%), Small Münsterländers (3/200, 1.5%), Tibetan Mastiffs (3/200, 1.5%), and 46 (23%) more from 23 other breeds. There were 76 (38%) males and 124 (62%) females.

The inclusion criteria were VDHE views performed for the HD scoring of dogs older than 12 months, with adequate technical quality in terms of image and positioning for HD scoring. Due to the observational nature of the study, owner consent and ethical committee approval were waived.

### 2.1. Radiographic Measurements

The 400 hip joints were scored into five categories for HD by M.G. using FCI criteria and a DICOM viewer and image analysis software (Dys4Vet version 1.1.0, accessed 12 June 2022). The criteria were as follows: A (no signs of HD—Norberg angle (NA) around 105° or more and excellent congruency); B (transitional or borderline hip joint—NA around 105° and mild incongruency); C (slight HD—NA around 100°, centre of femoral head outside of dorsal acetabular margin and moderate incongruency); D (moderate HD—NA around or greater than 90°, signs of osteoarthritis and obvious or considerable incongruency); E (severe HD—NA lower than 90°, signs of osteoarthritis and severe incongruency) [2,4]. The NA was measured between a line that joins the center of the femoral heads, and another line connecting each center of the femoral head with the cranial effective acetabular rim (Figure 1) [17]. Hip joints with NA around 105° or greater were considered normal. When the NA was around 100° or lower they were scored with different degrees of dysplasia [4].

The acetabulum and the proximal femur were delimitated using the image polygonal annotation tool (LabelMe version 4.5.13 accessed between 1 January and 31 March 2022) [16]. The acetabular area (AA) and the acetabular area occupied by the femoral head (AAOFH) were measured and the HCI was calculated by dividing AAOFH by AA (Figure 2).

In a preliminary study, the HCI measurements were performed by two examiners, D.M. (E1) and P.F. (E2), in 25 dogs (50 hip joints) and in two independent sessions (S1 and S2), to study intra- and inter-examiner reliability and agreement. Then, in the main study, each examiner measured the HCI in their randomly assigned half of the sample (100 dogs per examiner). The FCI scoring and the HCI measurements were performed in a double-blind fashion.

### 2.2. Statistical Analysis

Statistical analysis was performed using the computer software SPSS (SPSS Statistics for Windows Version 27.0: IBM Corp., Armonk, NY, USA). The data analysis was performed on joints individually. A *p*-value of < 0.05 was considered statistically significant.

For the preliminary study, the Central Limit Theorem was adopted, which states that for sufficiently large samples sizes (*n* > 30) the distribution of the mean for a variable tends to be normally distributed, independently of the distribution of the population from where that variable originated, and so parametric tests for data analysis were used [18]. The paired *t*-test was used to determine if there was a systematic difference between the duplicate measurements of S1 and S2 of each examiner, as well as between examiners E1 and E2 [19]. Additionally, the Bland–Altman analysis was used to offer insight into the pattern and extent of agreement. The 95% limits of agreement (LOAs) were calculated as the mean difference ± 1.96 standard deviation (SD). When the 95% confidence interval (CI) of the mean difference included zero, measurements were considered to be in agreement; and when the 95% lower and upper LOA were small, measurements were considered to be equivalent [20,21]. The intraclass correlation coefficient (ICC_3,1_ model) was used to evaluate the intra- and inter-examiner reliability [19,22]. ICCs of 0, less than 0.5, between 0.5 and 0.75, between 0.75 and 0.9, greater than 0.9 and 1 were interpreted as random, poor, moderate, good, excellent and perfect reliability, respectively [19]. A lower limit 95% CI of ICC > 0.75 was defined as adequate reliability [22,23].

The normality and homogeneity of variance were verified, and the Welch’s ANOVA, followed by the post-hoc Games–Howell test, was used to compare mean HCI values between FCI categories. The null hypothesis was that there were no significant differences in the HCI mean values between the FCI categories [24].

## 3. Results

In the preliminary study, the HCI mean difference ± SD between Examiner 1 sessions (E1S1 and E1S2) was 0.005 ± 0.027 (95% CI [−0.003, 0.013], *p* = 0.178, in paired *t*-test) and in Examiner 2 sessions (E2S1 and E2S2) was 0.002 ± 0.014 (95% CI [−0.002, 0.006], *p* = 0.282, in paired *t*-test). The HCI mean difference ± SD between Examiner 1 and Examiner 2 (E1S2 and E2S2) was 0.005 ± 0.025 (95% CI [−0.002, 0.012], *p* = 0.147, in paired *t*-test) (Figure 3, Figure 4 and Figure 5).

The ICC for HCI between E1S1 and E1S2 was 0.968 (95% CI [0.944, 0.981], *p* < 0.001), between E2S1 and E2S2 it was 0.992 (95% CI [0.986, 0.996], *p* < 0.001) and between E1S2 and E2S2 it was 0.972 (95% CI [0.952, 0.984], *p* < 0.001).

A total of 120 (30%) hip joints were scored as FCI grade A, the HCI mean ± SD was 0.739 ± 0.044; 157 (39.25%) hip joints were scored as FCI grade B, the HCI mean ± SD was 0.666 ± 0.052; 68 (17%) hip joints were scored as FCI grade C, the HCI mean ± SD was 0.605 ± 0.055; 38 (9.5%) hip joints were scored as FCI grade D, the HCI mean ± SD was 0.494 ± 0.070; 17 (4.25%) hip joints were scored as FCI grade E, the HCI mean ± SD was 0.374 ± 0.122. The Shapiro–Wilk test indicated that all independent group samples were normally distributed (*p*_A_ = 0.917; *p*_B_ = 0.242; *p*_C_ = 0.563; *p*_D_ = 0.308; *p*_E_ = 0.193) and Levene’s test indicated unequal variances among group samples (*p* < 0.001). Welch’s ANOVA followed by the post hoc Games–Howell test indicated that the means of all five independent groups differed significantly from each other (*p* < 0.01) (Table 1, Figure 6).

## 4. Discussion

In this work, we studied the relationship between the acetabular coverage area and the femoral head, calculating the HCI and its variation with the FCI categories. Image semantic segmentation, one of the most pivotal approaches in medical imaging [25], was applied to digital radiographic images to isolate the proximal femur and acetabulum from the background, and to identify any shared (overlay) pixels. The HCI seems to be a suitable variable for measuring hip congruency, since it is directly related to the free space present in the hip joint socket. The perfect hip congruency is associated with an even, parallel joint surface that translates into reduced acetabular free space, characteristic of normal hips, which tends to increase with hip subluxation and becomes maximum in hip dislocation [2,4]. FCI hip scoring categories A, B, C, D and E are directly related to excellent congruency; transition to mild; moderate; considerable or severe incongruency, respectively [2,6,7]. The FCI allows official reading of hips through visual image evaluation by a veterinarian scrutineer selected by local breed clubs; consequently, the quality of the scoring is subjective and may vary greatly [8]. This may justify the poor inter-observer agreement reported by some studies [9,11]. On the other hand, with respect to the FCI grading, NA measurement shows sufficient reproducibility, even among less-experienced examiners [26,27]. This prompts us to say that part of the scoring divergences can be imputed to the evaluation of the congruency parameter, based on description presented by the FCI. Congruency is valuable for the classification of HD into categories, especially to distinguish between animals with no signs of dysplasia/near-normal hip joints and animals with mild dysplasia, since there is some proximity and overlap in NA value among these hip categories. To some extent, the inaccurate assessment of hip congruency can lead to the exclusion of false positives from selective breeding, or the inclusion of false negatives with erroneous screening results. There is some variability in the literature about NA value interpretation, as different cut-points for NA have been reported, refuting the premise that the widely accepted 105° is the determinant NA value to distinguish between normal and abnormal hips [17,28,29]. Taking this into consideration, there is a perceived need to give greater weight to estimates of hip congruency and conformation to aid the NA method, and thus provide more reliable and objective HD scores.

Our results showed that the methodology behind the HCI had adequate intra- and inter-examiner reliability and agreement. Considering the results of the paired *t*-tests, there was no evidence of a systematic bias between the measurements of the examiners’ sessions, or between the two examiners, as the mean differences were ≤0.027 and all the 95% CIs included zero and were narrow. One would expect that the differences in the smaller HCI mean values would be larger than the differences in bigger mean values due to bone remodeling present in severe cases of HD, which could have increased the segmentation difficulty and caused more disparity between sessions and examiners. However, analyzing all three Bland–Altman diagrams, we can observe that the differences in values do not increase or decrease in proportion to the average values, so there is no evidence of proportional bias. Further analysis of the diagrams enables us to say that E1 consistently measured slightly higher than E2. Finally, the 95% limits of agreement may be considered clinically small, so the methodology practiced by the two examiners can be used interchangeably [20]. The intra-examiner ICC was ≥0.967, whereas the inter-examiner ICC was 0.972. All lower limits of the 95% CIs of ICCs were ≥0.942; this is substantially higher than the adequate reliability stipulated at 0.75, which translates to excellent reliability. This demonstrates that the methodology and the method produce measurements that are repeatable and reproducible with ease. For this study, the examiners had roughly the same amount of practice time with the method, so inferences about different levels of experience are not applicable.

Our results indicate that different FCI categories A, B, C, D and E have HCI means that gradually decrease with statistically significant differences (*p* < 0.01) (Table 1; Figure 6). Therefore, the null hypothesis is rejected, and this supports our assertion that the HCI method is an objective parameter to differentiate FCI categories. The maximum HCI value of 1 can never be reached (the maximum in our study was 0.850), since even in cases of perfect congruency between the acetabulum and the femoral head there is always the radiolucent articular cartilage space and the femoral head fovea is not circular. Moreover, in the VDHE view a part of the femoral neck is also encompassed by the acetabulum, which is not spherical and does not conform to the contour of the acetabulum. However, the HCI should be quite sensitive to the cases in which acetabulum’s weight-bearing surface is deviated inwards, characteristic of the development of HD [30].

The index measurement method appears to have several strengths, as it does not use restrictive gauges to outline the oftentimes tortuous projected anatomy, but uses instead a software drawing tool that allows accurate demarcation of the sometimes complex outline of the subchondral bone of the structures concerned. The tool is especially useful to detail the morphology of the *margo acetabuli*. Furthermore, the HCI method can minimize the recognizable impact that positioning has on the projected anatomy [31,32], since the relative magnification of the relevant structures (acetabulum, femoral head and free space in between) in a single joint is similar, as they lie approximately in the same anatomical plane. As the HCI is an index, it compensates for variations in magnification, which allows for comparison between dogs. In addition, since it is a continuous variable, as opposed to the categorical scale of the FCI congruency, it enables comparison between ranges of values. The fact that our sample came from two different databases and included a wide variety of breeds should be interpreted as a positive feature of the study, as the results can be more easily extrapolated to canine populations susceptible to HD. However, some breeds with predisposition to develop HD are clearly overrepresented (Portuguese breeds) or underrepresented (Rottweiler and Tibetan Mastiff). In future, it would be desirable to carry out studies on a larger number of patients and breeds.

Like the NA method, HCI has some limitations and can never be used alone for FCI classification because of the existing overlap of HCI values between FCI categories. Some hips severely affected by osteoarthritis have their HCI value overestimated, because the fibrosis of the hip and the lack of depth of the acetabulum bring the femur closer to the acetabulum and reduce the free space, characteristic of animals with HD. Additionally, histological studies show that hips affected with severe osteoarthritis suffer from a significant modulation of trabecular bone structures that leads to increased bone volume and reduction of cartilage, specifically in the proximomedial area of the femoral head, corresponding to the main compressive region between femur and acetabulum [33,34]. However, we believe that, when used in conjunction with NA or other hip evaluation parameters, they can be complementary and very useful, especially in distinguishing animals without HD, with suspected HD or mild HD (A, B and C FCI categories, respectively). Therefore, each case has to be analyzed with caution, and all parameters should be considered in tandem. We see this predicament as a catalyst for future research to improve the HCI method, and to create a new quantitative method to objectively assess osteoarthritis.

## 5. Conclusions

The HCI shows satisfactory intra- and inter-examiner measurement agreement and reliability. The HCI means are gradually lower, with statistically significant differences between A, B, C, D and E FCI categories. The HCI measurement shows potential to make FCI classification more objective if incorporated as a classification criteria.

## Figures and Tables

**Figure 1 animals-12-02201-f001:**
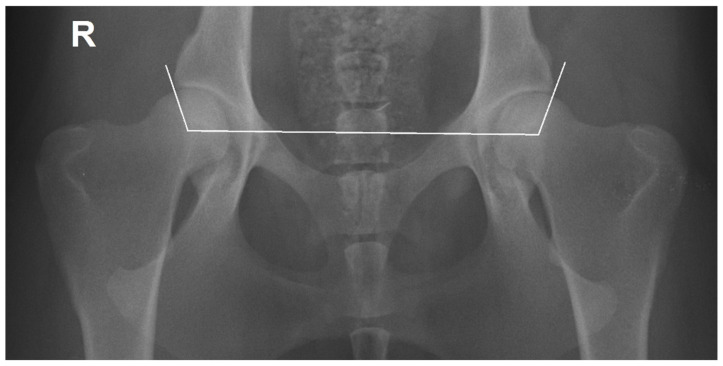
Ventrodorsal hip extended view of a female Transmontano Mastiff with no signs of hip dysplasia, right Norberg angle of 106° (left 110°) and an excellent congruency. The Norberg angle was calculated between the line joining the femoral centres of femoral heads and the other line connecting the centre of the femoral head and the cranial effective acetabular rim. R—right side.

**Figure 2 animals-12-02201-f002:**
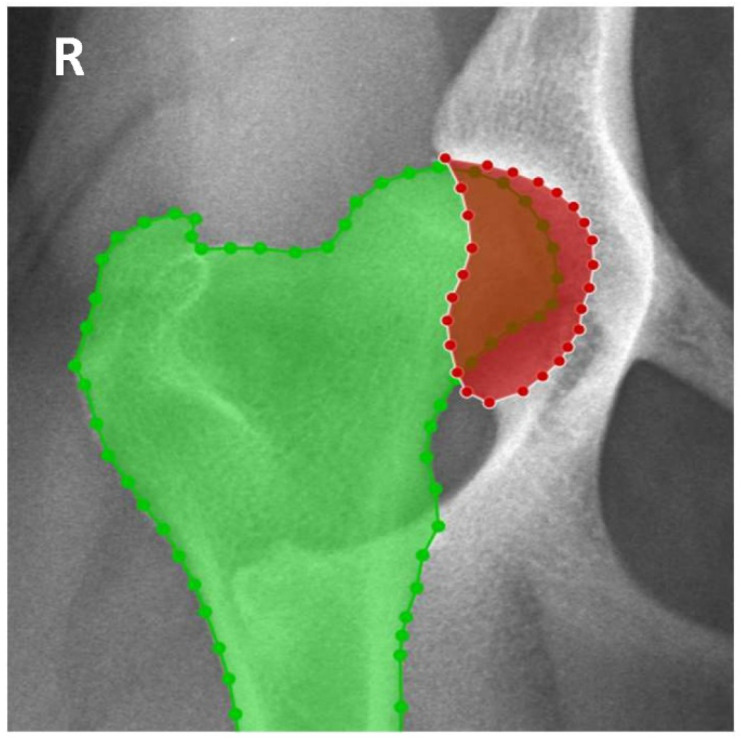
Delimitation of the proximal femur and acetabulum using the LabelMe annotation tool for the calculation of Hip Congruence Index (HCI). The acetabular area (AA) is delimitated by red points represented in red. The proximal femur is delimitated by green points and represented in green. The acetabular area occupied by the femoral head (AAOFH) is represented by the overlap between the red and green areas (HCI = AAOFH/AA). R—right side.

**Figure 3 animals-12-02201-f003:**
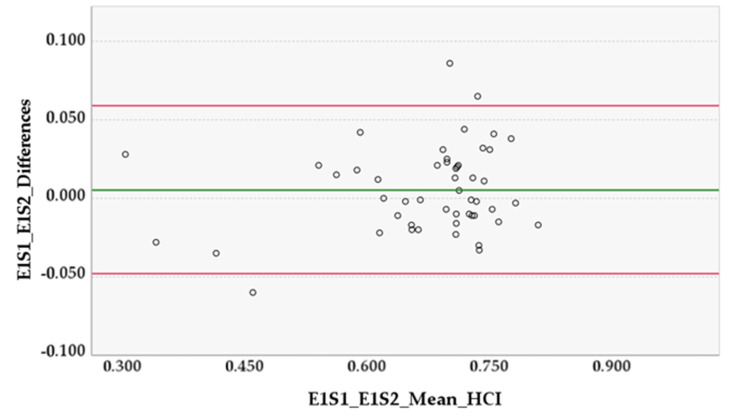
Differences between Examiner 1, Sessions 1 and 2 (E1S1 and E1S2). The horizontal lines represent the mean of the differences (0.005) and the upper and lower 95% limits of agreement, approximately 0.059 and −0.048, respectively.

**Figure 4 animals-12-02201-f004:**
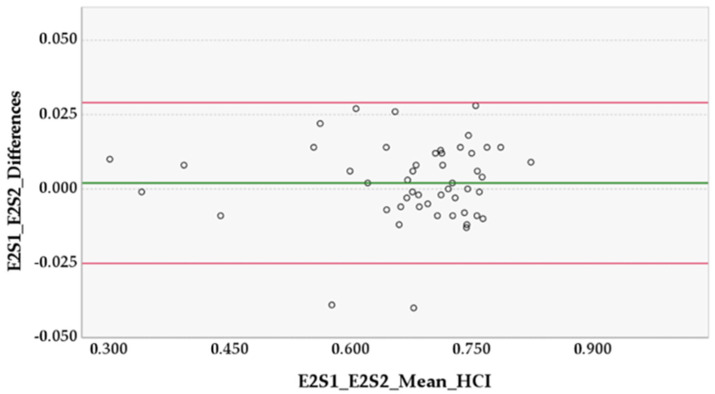
Differences between Examiner 2, Sessions 1 and 2 (E2S1 and E2S2). The horizontal lines represent the mean of the differences (0.002) and the upper and lower 95% limits of agreement, approximately 0.029 and −0.025, respectively.

**Figure 5 animals-12-02201-f005:**
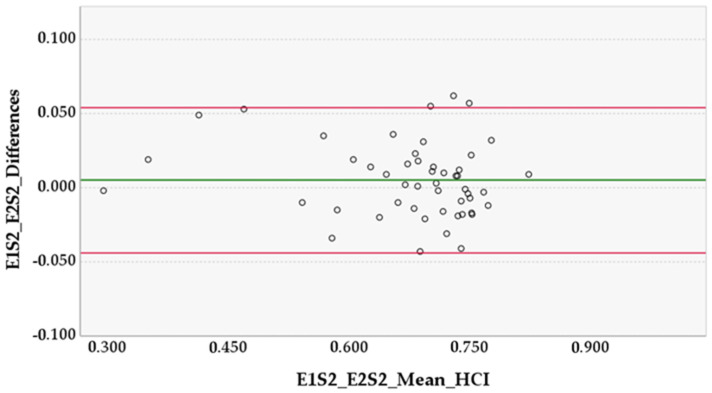
Differences between Examiner 1 and Examiner 2 in Session 2 (E1S2 and E2S2). The horizontal lines represent the mean of the differences (0.005) and the upper and lower 95% limits of agreement, approximately 0.054 and −0.044, respectively.

**Figure 6 animals-12-02201-f006:**
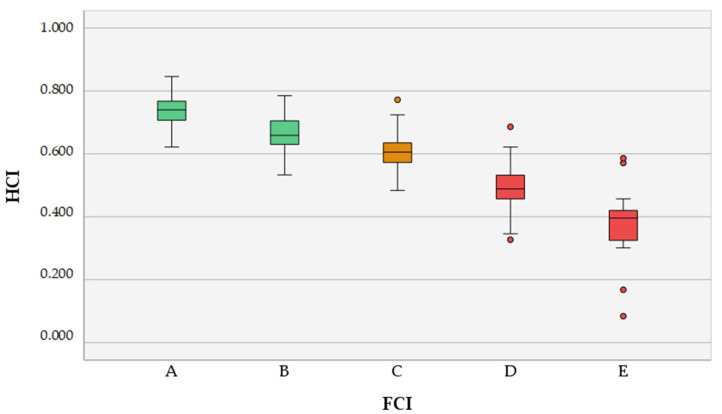
Box-and-whisker plot presenting the Hip Congruency Index classified in *Féderátion Cynologique Internationale* categories (A–E).

**Table 1 animals-12-02201-t001:** Statistic descriptive analysis of the Hip Congruency Index by *Fédération Cynologique Internationale* categories.

FCI Categories	*n*	Mean *	SD	Mean 95% CI		Min	Max
				Lower Bound	Upper Bound		
A	120	0.739 ^a^	0.044	0.731	0.747	0.622	0.846
B	157	0.666 ^b^	0.052	0.658	0.675	0.533	0.785
C	68	0.605 ^c^	0.055	0.591	0.618	0.484	0.772
D	38	0.494 ^d^	0.070	0.471	0.517	0.327	0.686
E	17	0.374 ^e^	0.122	0.311	0.437	0.084	0.586

* Means with different superscripts are statistically different (*p* < 0.01) in the post hoc Games–Howell test that followed Welch’s ANOVA. ^a,b,c,d,e^: there are superscripts, all the means have different letters so all are statistically different.

## Data Availability

The data presented in this study are available on request from the corresponding author.

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
