# Peer review of "Acetabular Coverage Area Occupied by the Femoral Head as an Indicator of Hip Congruency"

_animals, 2022, doi:10.3390/ani12172201_

Round 1

Reviewer 1 Report

The study addresses the ever-present problem of hip dysplasia in dogs. The paper is very interesting. Its significant drawback is that it is very unrepresentative of certain breeds such as the Rottweiler or Tibetan Mastiff. It would be desirable to carry out future studies on a larger number of patients.
[87] please add a short statement about the need for a dysplasia test to qualify the dog for breeding.
[121] please add a diagram or figure of an X-ray with a measurement of the Norberg account

Author Response

Your Comment (YC): The study addresses the ever-present problem of hip dysplasia in dogs. The paper is very interesting. Its significant drawback is that it is very unrepresentative of certain breeds such as the Rottweiler or Tibetan Mastiff. It would be desirable to carry out future studies on a larger number of patients. Our answer (OA): Thank you very much for your comments and for the similarities in some of our thoughts about hip dysplasia. Some aspects about the breeds in the sample were added
YC: [87] please add a short statement about the need for a dysplasia test to qualify the dog for breeding. OA: Added
YC: [121] please add a diagram or figure of an X-ray with a measurement of the Norberg account OA: Added

Reviewer 2 Report

This study is interesting and well done.

1. What is the main question addressed by the research? 

The main question is to verify if there is a relationship between the acetabular coverage area and the femoral head and its association with the level of hip congruency.

2. Do you consider the topic original or relevant in the field, and if
so, why?

yes, because they describe a new possible method to evaluate the hip congruency

3. What does it add to the subject area compared with other published
material?

it is a new method

4. What specific improvements could the authors consider regarding the
methodology?

  It is a new methodology, it could be interesting    

5. Are the conclusions consistent with the evidence and arguments presented, and do they address the main question posed?  

yes   

6. Are the references appropriate?  

yes  

7. Please include any additional comments on the tables and figures.  

The tables are well done and help in understanding the text

Author Response

Thank you very much for your positive comments and for the similarities in some of our thoughts about hip dysplasia. 

Reviewer 3 Report

Dear Authors

The manuscript submitted for review tackles topics that are topical and important not only for the animal species in question. The work is written in comprehensible language, even for a non-specialist, and, above all, will be interesting reading for breeders of the described species. I have some comments that in my opinion will improve the quality of the information presented.  In the Simple Summary section, please explain the abbreviation NA. The Abstract section should be shortened and rewritten, as the authors give too much detail here, which is dedicated especially to the Materials and Methods section. Introduction-please supplement this section with information regarding the dog breeds in which hip dysplasia is most common, and those dog breeds in which the problem of hip dysplasia is rare.  The hip joint especially in the dog and cat has a greater range and versatility of motion than in other domestic animal species (the particular ability to invert the limb is confirmed by the ease with which the dog lifts the limb when urinating). So please also, in this section, refer to other domesticated animal species. Is hip dysplasia a common problem due to the design of the hip joint in these species? Materials and Methods-explain to readers the concept of Norberg Angle-between which parts of the hip joint does it occur? What is its normal range, and what range indicates displacement or suggests dysplasia? Fig. 1 -should be described in detail for readers. The results presented are interesting, including the discussion, so it would be good to complete the Conclusions section based on them.

Regards

Author Response

Your comment (YC): The manuscript submitted for review tackles topics that are topical and important not only for the animal species in question. The work is written in comprehensible language, even for a non-specialist, and, above all, will be interesting reading for breeders of the described species. I have some comments that in my opinion will improve the quality of the information presented.  Our answer (OA): Thank you very much for your positive comments.

YC:  In the Simple Summary section, please explain the abbreviation NA. OA: Added.

YC: The Abstract section should be shortened and rewritten, as the authors give too much detail here, which is dedicated especially to the Materials and Methods section. OA: Rewritten

YC: Introduction-please supplement this section with information regarding the dog breeds in which hip dysplasia is most common, and those dog breeds in which the problem of hip dysplasia is rare.  The hip joint especially in the dog and cat has a greater range and versatility of motion than in other domestic animal species (the particular ability to invert the limb is confirmed by the ease with which the dog lifts the limb when urinating). So please also, in this section, refer to other domesticated animal species. Is hip dysplasia a common problem due to the design of the hip joint in these species? OA: Some information about dog hip biomechanics Added.

YC: Materials and Methods-explain to readers the concept of Norberg Angle-between which parts of the hip joint does it occur? What is its normal range, and what range indicates displacement or suggests dysplasia? OA: Indicated.

YC: Fig. 1 -should be described in detail for readers. OA: The figure legend was improved.

YC: The results presented are interesting, including the discussion, so it would be good to complete the Conclusions section based on them. OA: Conclusions were rewritten.